# Histone dynamics mediate DNA unwrapping and sliding in nucleosomes

Grigoriy A. Armeev [1,2,6], Anastasiia S. Kniazeva[1,6], Galina A. Komarova[3], Mikhail P. Kirpichnikov[1,4] & Alexey K. Shaytan [1,2,5✉]

Nucleosomes are elementary building blocks of chromatin in eukaryotes. They tightly wrap ~147 DNA base pairs around an octamer of histone proteins. How nucleosome structural dynamics affect genome functioning is not completely clear. Here we report all-atom molecular dynamics simulations of nucleosome core particles at a timescale of 15 microseconds. At this timescale, functional modes of nucleosome dynamics such as spontaneous nucleosomal DNA breathing, unwrapping, twisting, and sliding were observed. We identified atomistic mechanisms of these processes by analyzing the accompanying structural rearrangements of the histone octamer and histone-DNA contacts. Octamer dynamics and plasticity were found to enable DNA unwrapping and sliding. Through multi-scale modeling, we showed that nucleosomal DNA dynamics contribute to significant conformational variability of the chromatin fiber at the supranucleosomal level. Our study further supports mechanistic coupling between fine details of histone dynamics and chromatin functioning, provides a framework for understanding the effects of various chromatin modifications.

[1] Department of Biology, Lomonosov Moscow State University, Moscow, Russia. [2] Sirius University of Science and Technology, Sochi, Russia. [3] Department of Physics, Lomonosov Moscow State University, Moscow, Russia. [4] Shemyakin-Ovchinnikov Institute of Bioorganic Chemistry, Russian Academy of Sciences, Moscow, Russia. [5] Bioinformatics Lab, Faculty of Computer Science, HSE University, Moscow, Russia. [6]These authors contributed equally: Grigoriy A. Armeev, Anastasiia S. Kniazeva. ✉email: shaytan_ak@mail.bio.msu.ru

Many chemical and physical signals regulate the processing of genetic information in living cells. In eukaryotes, this regulation largely happens at the level of nucleosomes—repeating units wrapping DNA around octamers of histone proteins[1,2]. Four types of histones (H3, H4, H2A, H2B) form two types of dimers (H3-H4 and H2A-H2B). Four of these dimers form an octamer with a twofold symmetry axis (the dyad axis)[3]. As seen in crystallographic studies, ~147 DNA base pairs bend sharply around an octamer in ~1.7 turns of a left-handed superhelix forming a nucleosome core particle (NCP)[4] (Fig. 1a). Flexible histone tails protrude from the globular part of the octamer[5]. The stability of nucleosomes can vary by 2–4 kcal mol$^{-1}$ depending on the DNA sequence bendability[6], which was shown to depend on the GC content, the presence of flexible YR dinucleotide steps, poly(dA:dT) tracts, epigenetic modifications of DNA bases, such as 5-methylcytosine, etc[7–10]. Variation of histone sequence is another important factor. Despite being very conserved through evolution, histone proteins are subject to many functionally relevant variations, including posttranslational modifications (PTMs)[11] and sequence alterations through the incorporation of histone variants, isoforms, and mutations[12–14].

The view of nucleosomes as mere DNA compaction vehicles has now become obsolete. Various modes of ATP-dependent and ATP-independent nucleosome dynamics modulated by histone and DNA composition provide a rich, dynamic landscape for genome regulation[15]. For example, the sliding of nucleosomes by ATP-dependent chromatin remodelers[16] or their reconfiguration by RNA polymerases[17,18] is heavily involved in transcription and its regulation. Passive nucleosome dynamics are also implicated in many processes. Pioneering experiments by Jon Widom showed that transcription factors may exploit transient DNA unwrapping to access their binding sites[19]. Recent FRET[20,21], NMR[22,23], cryo-EM[24,25], mass spectrometry, and cross-linking studies[22,26,27] suggest that other, more nuanced dynamical modes exist that are sensed and exploited by chromatin proteins. For example, DNA twist-defects within NCP provide a pathway for ATP-dependent nucleosome remodeling[24,28]. The intrinsic dynamics (plasticity) of the histone core has also been implicated in this process (although not without some debate)[22,29,30]. Suppressing these dynamics by the introduction of site-specific disulfide cross-links into individual H3–H4 dimers was reported to inhibit nucleosome sliding by SNF2h, increase octamer eviction by RSC[22], block thermally induced diffusion of nucleosomes

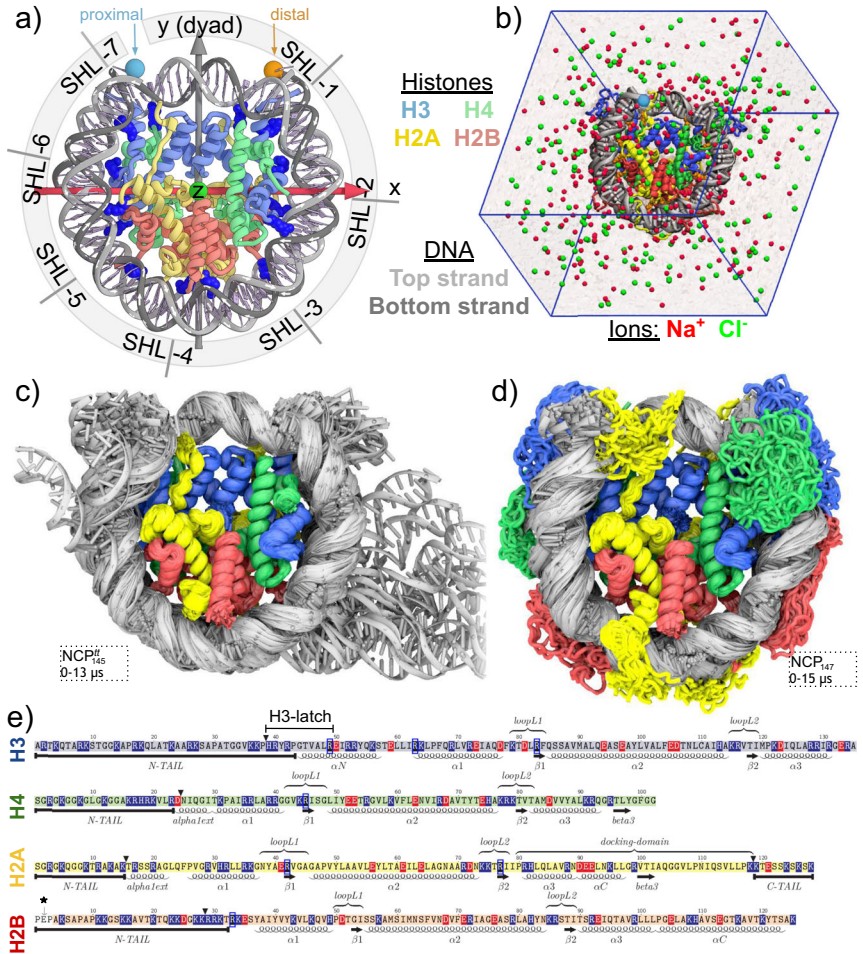

**Fig. 1 Overview of NCP structure and dynamics. a** NCP and its reference axes (z - superhelical, y - dyad). White arrows on DNA strands show 5′-3″ direction. Spheres highlight the proximal and distal ends of the double helix. Superhelical locations (SHL) are shown for the proximal half of the DNA (SHL < 0). Proximal H3-H4/H2A-H2B dimers are in the front (z > 0). Key arginines inserted in DNA minor grooves are shown in dark blue. **b** NCP in a simulation box with solvent. **c, d** Dynamics of NCP with truncated and with full-length histone tails (NCP$^{tt}_{145}$ and NCP$_{147}$ systems, respectively). Overlay of MD snapshots spaced 0.1 μs apart. **e** Sequences of the core histones and their secondary structure features: α-helices, β-strands, loops, flexible histone tails, etc. For simulations with truncated histone tails, the respective positions are marked with black triangle. Positively and negatively charged residues are highlighted in blue and red, respectively. Key arginines are highlighted with dark blue frames. Black asterisk - H2B residues absent in the recombinant protein.

along the DNA[25] and even impair compaction of chromatin by HP1 protein[27]. These results suggest that histone octamer may adopt conformations alternative to those seen in X-ray structures. Indeed using advanced cryo-EM techniques Bilokapic et al. have recently observed some alternative nucleosome conformations with unwrapped DNA, tilted histone α-helices, and overall squeezed shape of the NCP[25,31]. Even bigger internal fluctuations within the histone octamer structure are suggested by lysine cross-linking experiments[26] and high-resolution FRET experiments[20].

These and other studies have highlighted the existence of a new level of dynamic complexity in nucleosomes with an emphasis on atomistic details. However, the structural interpretation of these results still remains elusive. In silico approaches, such as molecular dynamics (MD) simulations, are powerful tools that may supplement experimental methods and help to interpret their results mechanistically with a high level of detail. So far, MD simulations have been applied to study counterion and hydration patterns around nucleosomes[32], details of DNA dynamics, unwrapping, nucleosome sliding and disassembly[33–39], dynamics of histone tails[40–43], effects of histone PTMs[44–46], histone variants[47] and DNA sequence[48] on nucleosome dynamics, interactions between oligonucleosomes[49] and histone H1[50,51], etc. Due to computational complexity, insights are still limited either by the approximations of the model (e.g., removing histone tails, using implicit solvent models, or coarse-grained representation of the molecular system) or the achievable simulation timescales. Currently, the best-reported MD studies of nucleosomes in the all-atom representation have been limited to several microseconds of simulation time[37,42], even with the application of specialized supercomputers, such as ANTON. However, important functional transitions in nucleosome structure are expected to happen at the microsecond to millisecond timescale[20]. New functional dynamics modes at these timescales are still awaiting detailed structural characterization with MD simulations.

In this work, motivated by accumulating experimental evidence for previously unknown modes of nucleosome conformational dynamics and plasticity, we aimed at systematically exploring equilibrium nucleosome dynamics with all-atom MD simulations at a 15 microsecond timescale. We obtained long dynamical trajectories for maximally realistic atomistic models of NCPs with full-length histone tails at physiological temperature and ionic strength. Through specially designed trajectory analysis algorithms (coordinate projection in the nucleosome reference frame (NRF), DNA relative twist calculations, stable contacts analysis, etc.), we revealed functionally important modes of nucleosome dynamics and plasticity. We observed reconfiguration and unwrapping of DNA ends in nucleosomes mediated by H3 and H2A histone tails, the propagation of twist-defects in nucleosomal DNA, and structural plasticity of the histone core consistent with recent experimental observations. The implication of our results for understanding nucleosome sliding, remodeling, and higher-order chromatin structure organization are analyzed.

## Results

**Simulations overview.** We have performed long (up to 15 μs) continuous all-atom MD simulations of NCPs in explicit solvent (Fig. 1a, b). The set of simulated systems was based on high-resolution NCP X-ray structures from the PDB database. Systems' names and parameters are provided in Table 1 and Supplementary Table 1. $NCP_{147}$ is the main system in our analysis based on the highest available resolution X-ray structure. It harbors 147 bp of nucleosomal DNA that is split by the central dyad base pair into two strictly palindromic 73 bp halves. The pseudo-symmetry of the whole system (DNA and histones) enables combined analysis of the symmetry-related halves of the NCP and attribution of any differences to thermodynamic fluctuations. The set of simulated systems with truncated histone tails ($NCP^{tt}_{147}$, $NCP^{tt}_{145}$, $NCP^{tt}_{146}$) allowed us to simultaneously probe DNA dynamics not restricted by histone tails' binding and stability of different DNA conformations observed in X-ray structures. The latter are known to differ by the position of DNA twist-defects located around superhelix locations (SHLs) ±2 or ±5, where DNA is stretched or compressed, allowing for the same superhelical path to be covered by 145, 146 or 147 DNA base pairs (Supplementary Fig. 1).

Upon visual inspection of the obtained MD trajectories (Fig. 1c, d, Supplementary Fig. 2, Supplementary Movies 1–5), it became apparent that the 15 μs timescale covered by our simulations enabled us to observe dynamical modes that did not manifest at the timescale of only several microseconds. The most conspicuous mode was the detachment and unwrapping of the DNA ends from the histone octamer. The DNA detachment was more pronounced for the systems with truncated histone tails, consistent with experimental studies of the histone tail cleavage effects[52]. However, even for these systems, depending on the simulation run, one needed to wait from 1 to 6 μs before such dynamics were observed. More nuanced changes included changes in the translational register of the DNA strands, variation in DNA twisting, local DNA deformations, movement of histone helices, and histone tails rearrangements. To formally analyze and characterize these dynamical modes, we developed and applied specific mathematical approaches (see next sections).

We believe that open sharing and the ability to interactively view the MD trajectories by the readers greatly enhances the value of our results for the community. For this purpose, we provide a web interface hosted through a GitHub repository available at https://intbio.github.io/Armeev_et_al_2021. This interface allows one to interactively look through the MD trajectories via NGL viewer JavaScript library[53] and observe unwrapping values via interactive charts.

**Table 1 Simulated systems.**

| Name | PDB ID[a] | Description | Time, μs |
|---|---|---|---|
| $NCP_{147}$ | 1KX5 | 147 bp quasi-palindromic α-satellite DNA in pseudo-symmetric conformation, full-length histone tails in symmetric starting positions | 15.0 |
| $NCP^{tt}_{147}$ | 1KX5 | Same as $NCP^{147}$, but with truncated histone tails[b] | 10.0 |
| $NCP^{tt}_{145}$ | 3LZ0 | 145 bp Widom 601 DNA sequence, truncated histone tails[b] | 15.0 |
| $NCP^{tt}_{146}$ | 1AOI | 146 bp palindromic α-satellite DNA sequence, truncated histone tails[b] | 8.0 |
| $NCP^{fixed}_{147}$ | 1KX5 | Same as $NCP_{147}$, but Cα-atoms of histone folds (α1, α2, α3-helices) were restrained in motion | 8.0 |

[a]PDB database ID used to derive the simulated system model.
[b]See Fig. 1e for location of truncation sites.

**Analysis of DNA-histone interactions**. We start by establishing a framework to analyze DNA-histone interactions, which in turn will be used to quantify nucleosome dynamics. As a reference, we sought to extract the pattern of key DNA-histone interactions along the DNA as seen in the canonical NCP X-ray structure. However, X-ray structure alone does not allow to distinguish between important stable interactions and transient ones, such as those formed by histone tails. MD simulations provide a solution. We took the first 1000 snapshots from NCP$_{147}$ simulation (collected during the first microsecond of the simulation, during which the nucleosomal DNA retained its crystal-like conformation) and for every nucleotide identified amino acid residues that were in contact with them in at least 90% of MD frames. Since in NCP$_{147}$ DNA composition and initial conformation were pseudo-symmetric between the two halves of the NCP, as an additional requirement for our analysis, we selected only nucleotide-residue contacts present on both sides of the NCP. The resulting profile is shown in Fig. 2a (Supplementary Fig. 3 provides a view along the histone sequence at the atom-atom contact level). The residues forming stable contacts with DNA, termed anchor residues, form around 15% of all atom-atom contacts between DNA and histones, but delineate the topology of DNA-histone interactions in the canonical nucleosome structure. Monitoring the changes in anchor residues' contacts with nucleosomal DNA along the full MD trajectory allows characterizing rearrangements of histone-DNA interactions due to DNA unwrapping, sliding, or DNA

distortions (Supplementary Movies 6–10). Several aspects of stable interactions' pattern are particularly interesting.

First, there is an asymmetry in the number of contacts between the two complementary DNA strands, 12 for the top strand vs. 36 for the bottom strand. Second, the first half turn of the DNA coordinated by the H2A-H2B dimer and H3 αN helix (positions from −73 to −32 in Fig. 2a) has much less stable nucleotide-residue interactions than the inner half-turn of the DNA coordinated by the H3-H4 dimer (positions from −36 to 0) (17 vs. 31 stable contacts). This is consistent with the easier unwrapping of the outer turns of nucleosomal DNA observed in DNA pulling experiments[54]. A significant contribution to this surplus (9 interactions) comes not from the histone folds but from the H3 αN-helix and the nearby segment of the H3 histone tail. Third, it can be seen on the profile that the portion of the H3 tail adjacent to the H3 αN-helix has a unique property—three residues H3Y41, H3R42, and H3T45 form simultaneously stable interactions between position -9 close to the dyad and position 71 at the end of the nucleosomal DNA. Analysis of individual atom-atom contacts between histones and DNA (Supplementary Fig. 3) and visual analysis suggests that a larger region of histone H3 between residues H3H39 and H3R49 acts as a latch by holding the nucleosomal DNA ends attached to the underlying DNA gyre. We refer to this region further as "H3-latch" (Fig. 1e).

Analysis of the total number of histone-DNA atom-atom contacts and its variation with time provides another way to

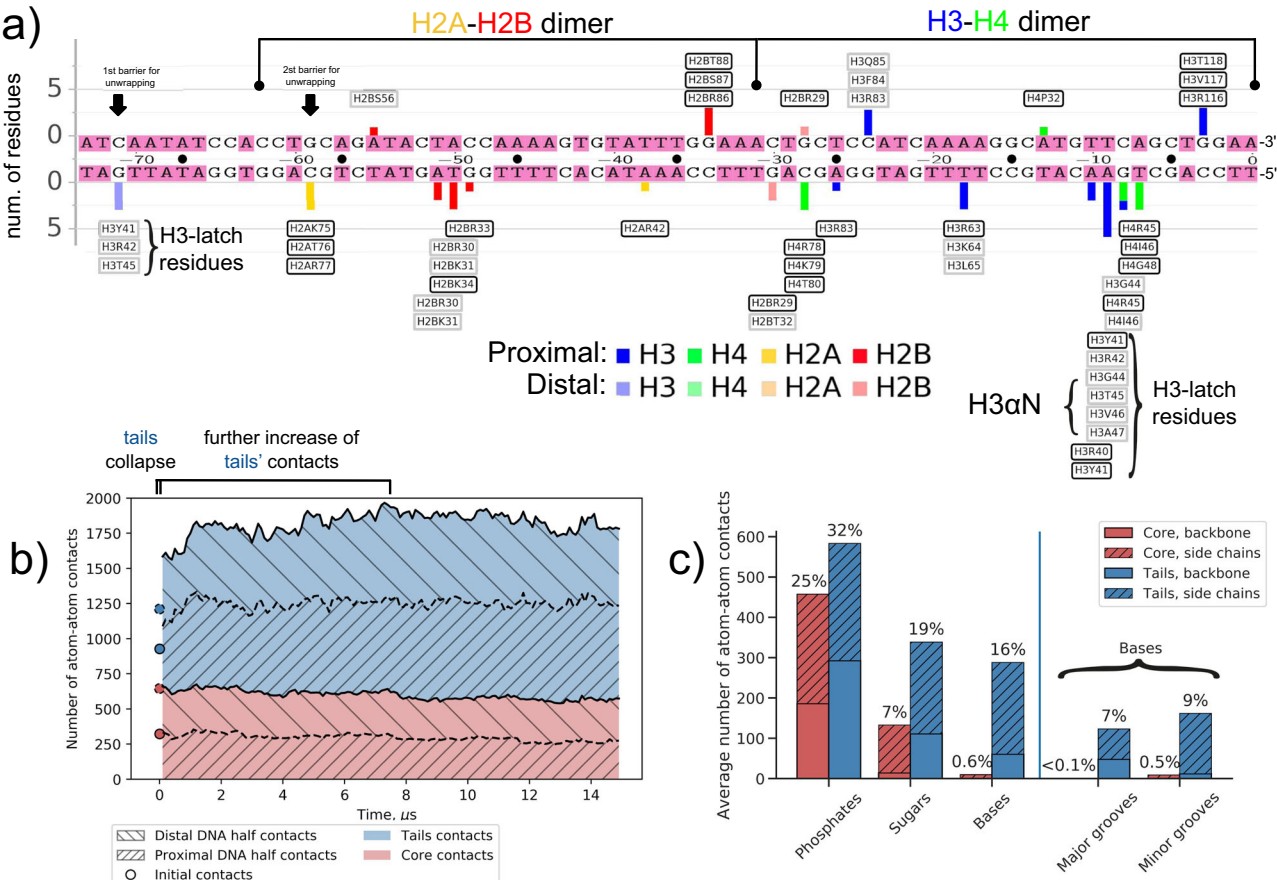

**Fig. 2 DNA-histone interactions in the nucleosome (based on NCP$_{147}$ simulation). a** Profile of stable amino acid residue-nucleotide contacts during the first microsecond of simulations plotted for the left half of the nucleosomal DNA. Individual histone residues are labeled on top of each bar. Residues that formed stable contacts for the entire 15 μs trajectory are shown in black frames. **b** Changes in the number of DNA-histone atom-atom contacts along MD trajectory. **c** The average number of histone-DNA atom-atom contacts classified by interacting entities: histone core or tail parts, DNA phosphates, sugars or bases.

quantify interactions (Fig. 2b, c, Supplementary Table 2, Supplementary Fig. 4). Positively charged histone tails tightly interact with negatively charged DNA and, on average, make ~67% of atom-atom contacts with the DNA. The majority of DNA-histone contacts are made with phosphate groups (~57%) and sugars (~26%). In accordance with the largely non-specific binding of DNA by histones, <1% of contacts were observed between the globular histone core and DNA bases (Fig. 2c). However, around 16% of the total number of contacts are made by the flexible histone tails with the DNA bases' atoms. Contrary to the predisposition of transcription factors to interact with the DNA major groove[55], histone tails make more interactions with the DNA minor grooves. A more detailed inspection revealed that minor groove interactions mainly stem from the insertion of arginine and lysine residues' side chains (Supplementary Fig. 4b). Arginines have a strong preference for minor groove binding, while lysines equally interact with major and minor grooves.

The number of DNA-histone contacts in NCP$_{147}$ system is presented in Fig. 2b as a function of simulation time. The collapse of initially partially extended histone tails onto the DNA happens rapidly within a few dozens of nanoseconds, consistent with earlier observations[41]. Our simulations revealed further much slower microsecond scale changes of histone-DNA interactions. At the multi-microsecond timescale, histone tails may explore and find new conformations that provide additional contacts with the DNA. A considerable increase of up to 37% in atom-atom contacts between histone tails and the DNA can be seen within the first 8 μs of simulations. The accommodation of the long H3 N-tail between the DNA gyres and the establishment of interactions between the H2A C-tail and the DNA ends contribute to this increase (Supplementary Fig. 4f). We also observed fluctuations in the number of contacts between the globular core part of histones and the DNA of around 10%. Changes in the histone and DNA conformations in different parts of the nucleosome, including the regions close to the dyad, contributed to these fluctuations (Supplementary Fig. 4g).

**DNA unwrapping.** Molecular details of spontaneous DNA unwrapping are one of the key results of our multi-microsecond simulations. We used two criteria to quantify the length of the unwrapped DNA segment, either a loss of contacts with the globular core (non-tail part) of the histone octamer (as defined in Fig. 1e) or a geometrical displacement of all the base pairs of the segment by more than 7 Å from the DNA path in the corresponding X-ray structure. The two approaches yielded comparable results with respect to the timing and magnitude of DNA unwrapping (Supplementary Fig. 5). In the further description, we rely on the geometric criterion. Following a reductionist approach, we start by analyzing DNA unwrapping in a system with truncated histone tails by the example of NCP$^{tt}_{145}$ as it presented a wider range of unwrapping modes (Fig. 3, Supplementary Fig. 6). We then discuss histone tails' effects on this process, as seen in NCP$_{147}$ simulation (Fig. 4, Supplementary Fig. 7).

An important conceptual framework for understanding nucleosomal DNA unwrapping emerges from the analysis of our simulations. DNA unwrapping states of NCP should not be characterized by the extent of DNA unwrapping per se (as is often presented schematically in literature) but rather should be characterized by the ranges of unwrapping values within which DNA rapidly fluctuates. For NCP$^{tt}_{145}$, as seen in Fig. 3 three main states were observed: 1 - the wrapped, X-ray like state characterized by the tight association of DNA with the histone octamer, 2 - the unwrapped state characterized by rapid fluctuations of DNA in the 0–15 bp unwrapping range, which

we refer to as "DNA breathing", 3 - DNA unwrapping of around 25 base pairs with fluctuations in the 18–28 bp range. The distinction between the notion of instantaneous DNA conformation and the dynamic state of DNA unwrapping is further illustrated by the fact that zero extent of DNA unwrapping which solely represents state 1 is often transiently encountered in state 2 also. Kinetic analysis (see Fig. 5) shows that for NCP without histone tails, it takes on average around 40 ns to get from a DNA conformation with 15 unwrapped base pairs to a fully wrapped conformation while transitions between different unwrapping states happen on microsecond timescale and beyond.

What triggers the transition between different unwrapping states? The examination of molecular interactions revealed that transition between state 1 and 2 is governed by the loss or reestablishment of interactions between DNA ends and a few residues of the H3-latch (Fig. 3, middle panel). H3H39, H3Y41, and H3R49 are inserted into the minor groove of the DNA end. In contrast, H3R40 is inserted into the minor groove of the neighboring DNA gyre and several other nearby residues, including H3P43, interact with its DNA backbone. DNA unwrapping requires the loss of histone contacts with the DNA end. Due to the flexibly of these residues' side chains, the reestablishment of these contacts presents a kinetic and entropic barrier separating the two states (see substates 1a and 2a in Fig. 3). Further destabilization of the region may include the detachment of the H3-latch from the underlying DNA gyre (substate 2b), which contributes to a longer lifetime of the unwrapped state. In NCP$^{tt}_{145}$ the rewrapping 2a → 1a transition took from 0.4 μs to 1.4 μs. The observed rewrapping time from substate 2b was around 8 μs. Interestingly an alternative conformation of the H3-latch was found (state 1b) that ensured the stably wrapped state but included the swapping of H3R40 and H3H39 positions in the minor grooves of the two neighboring DNA gyres.

The DNA unwrapping analysis in the system with full-length histone tails (NCP$_{147}$) is shown in Fig. 4. Several effects of histone tails on the dynamics of DNA unwrapping can be formulated from comparing with NCP$^{tt}_{145}$ simulation. First, during DNA detachment, histone tails continue to form many interactions with the DNA ends (Fig. 4, middle panel). Particularly, interactions are formed by the H3 N-tail and H2A C-tail. Even when the same number of DNA base pairs is detached, the spatial magnitude of the DNA fluctuations is suppressed relative to the systems with truncated histone tails because histone tails are holding the DNA close to the histone octamer (see Fig. 5a–d). These interactions also suppress further unwrapping of the DNA (state 3 was not observed). Second, histone tails' active interactions with the DNA ends make the kinetic and statistical landscape of DNA unwrapping more complex. State 2 is split into a series of dynamical substates that may persist on a multi-microsecond timescale. In NCP$_{147}$ simulation, we can discern at least two substates. In substate 2′ the terminal half-turn of the nucleosomal DNA rapidly fluctuates (0–7 bp). The H3-latch loses its interactions with the DNA, but the insertion of the H3 N-tail between the DNA gyres and the H2A C-tail into the minor groove of the DNA end from the inner side restricts further unwrapping (see Fig. 4). However, DNA still fluctuates rapidly and can reach the conformation of the fully wrapped state. Substate 2″ is characterized by the invasion of the N-terminus of histone H3 between the DNA and the surface of the octamer. The N-end of the H3 tail is effectively tucked between the DNA and the histone core at a position one helical turn away from the DNA end. This results in a state where DNA fluctuates in the unwrapping range of 4–13 bp and cannot reach its fully wrapped conformation. The state was observed for around 4 μs and resolved when the H3 N-terminus changed its position.

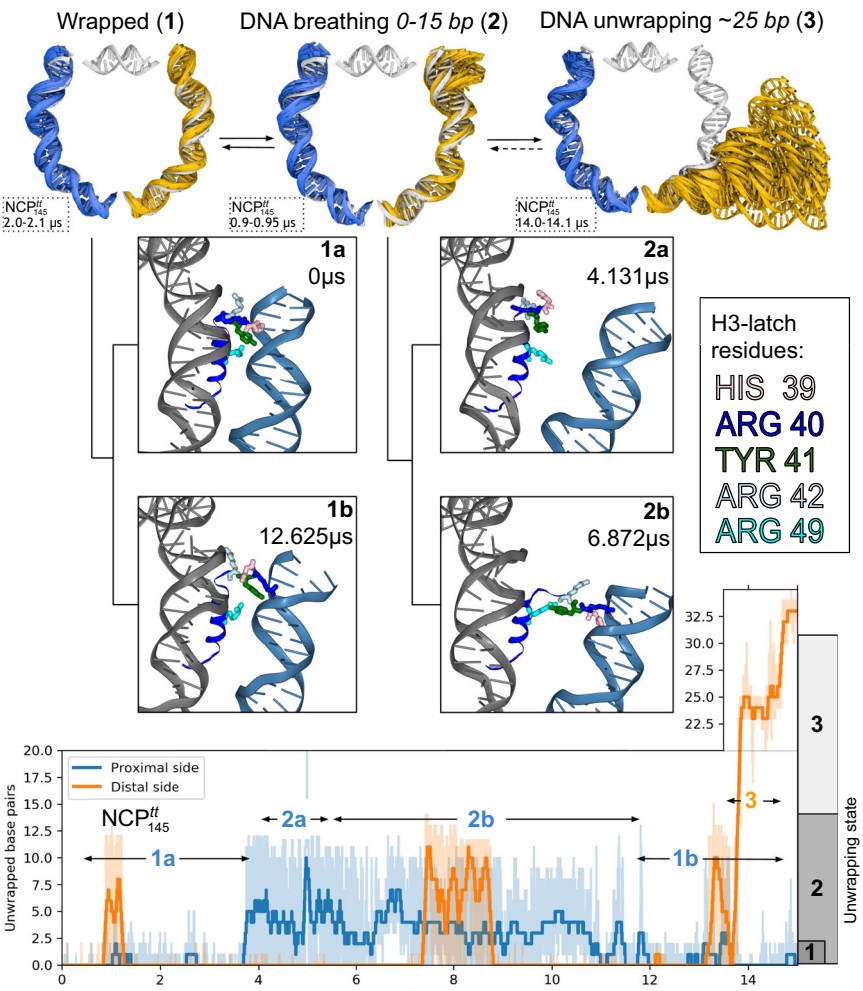

**Fig. 3 Mechanisms of DNA unwrapping in nucleosomes with truncated histone tails based on NCP$^{tt}_{145}$ simulation.** Top panel: different DNA unwrapping states. Middle panel: substates within each state showing different conformations of the H3-latch residues interacting with the nucleosomal DNA end and the neighboring DNA gyre. Bottom panel: profile of the extent of DNA unwrapping during MD simulations. Thin semitransparent lines are used to plot instantaneous unwrapping values; thick lines depict smoothed signal with Savitzky-Golay filter using 100 ns window and first-degree polynomial.

Collectively these findings highlight that histone tails change both kinetics and geometry of the DNA unwrapping.

To analyze the geometry of nucleosomal DNA in more detail, we plotted the DNA path (as defined by base pair centers) projected to NRF planes (Fig. 5a–d). The plot reveals that the unwrapping process happens not only in the plane of the nucleosome particle but also in an out-of-plane fashion. We did not observe a significant correlation between out-of-plane and in-plane displacement of the DNA end (Fig. 5e), suggesting that contrary to common perception, DNA unwrapping very often has both out-of-plane and in-plane components of comparable magnitude. Recently, Bilokapic et al. using cryo-EM analysis, were able to obtain structural subclasses of NCPs, including a class with bulged DNA[31]. Figure 5a–d show that such bulges lie well within conformational ensembles that we see in simulations.

To relate the magnitude of the observed DNA breathing motions to the data obtained by experimental FRET measurements, we estimated the distance fluctuations between FRET fluorophores' common attachment sites[56,57] and potential effects of these fluctuations on FRET efficiency (Supplementary Figs. 8, 9). Our analysis suggests that microsecond averaged variations in the distance between fluorescent labels may reach 1 nm due to

DNA breathing even for the system with full-length histone tails. While such magnitudes of spatial variation can be detected by FRET, the exact feasibility of the detection depends on the configuration of the labels (length of attachment linkers, Förster radius, etc.) and the temporal resolution of the measurements (see Discussion).

**DNA breathing facilitates chromatin fiber plasticity**. Given the observed equilibrium thermal fluctuations of nucleosomal DNA ends in the realistic NCP model with full-length histone tails (NCP$_{147}$), we sought to extrapolate these findings to chromatin structure and properties at the supranucleosomal level. A simple coarse-grained modeling approach (see Methods) was used to construct nucleosome fibrils by connecting random NCP conformations observed in MD by straight segments of 15 bp linker B-DNA. Connecting the X-ray conformation of NCP results in a well-known static "two-start" model of the chromatin fiber. Yet, accounting for DNA breathing reveals a very dynamic and polymorphic picture of chromatin fibrils already at the level of just several nucleosomes (Fig. 6a). In a hexanucleosome array, the average distance between the flanking nucleosomes increases from 23 nm for an ideal "two-start" fiber to around 30 nm,

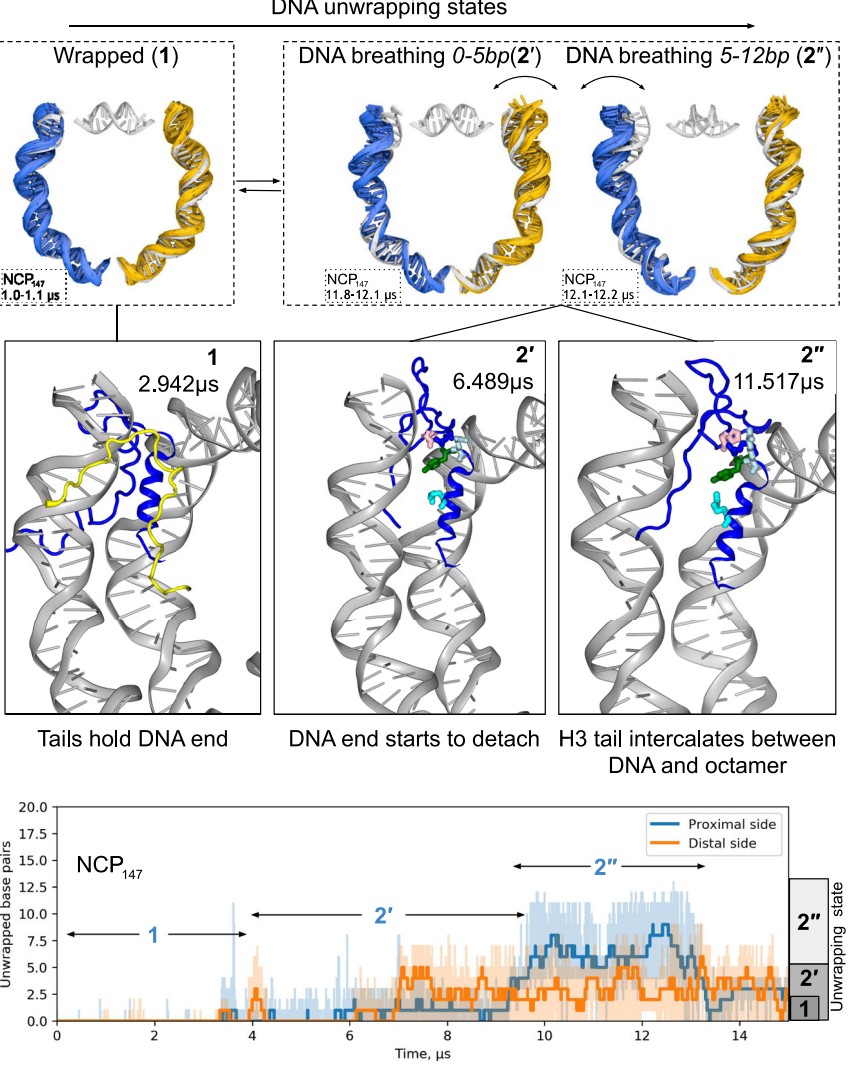

**Fig. 4 Mechanisms of DNA unwrapping in nucleosomes with full-length histone tails based on NCP$_{147}$ simulation.** Figure follows the design of Fig. 3. The middle panel shows typical conformations of histone tails and H3-latch residues for selected unwrapping states/substates.

reflecting the chromatin fiber's overall loosening due to thermal fluctuations. However, the distribution of end-to-end distances is broad (~13 nm FWHM). Modeling showed that DNA breathing alone can account for both small chromatin loops where flanking nucleosomes are in contact and structures twice longer than the ideal X-ray based fibril (Fig. 6a).

Simulations of longer chromatin fibers (up to 65 nucleosomes) (Fig. 6b) allowed us to estimate the decay of orientational correlations along the fiber and its persistence length solely due to DNA breathing. For the fibers based on NCP$_{147}$ system, we estimated persistence length to be around 16 nucleosome segments. We admit that the fiber's true persistence length will depend heavily on the internucleosomal interactions, which we do not account for in our model (except for steric clashes between NCPs). However, our model shows a fundamental mechanism of fiber flexibility and bendability due to nucleosomal DNA breathing. The amount of fiber bending enabled by this mechanism is likely to be on par with the contributions due to the bending of the linker DNA segments, which we did not account for in our modeling (the persistence length of straight B-DNA is around 150 bp, which amounts to ten nucleosome segments connected by 15 bp DNA linkers).

We repeated the above-described analysis for the NCP conformations taken from the NCP$^{tt}_{147}$ simulation to estimate the effects of histone tails' truncation and hence higher DNA breathing fluctuations and unwrapping angles. Predictably, this resulted in shorter persistent length (Fig. 6b, green vs. blue lines). The distribution of the end-to-end distances in hexanucleosome arrays became a bit wider, and its maximum shifted to shorter values. This is likely the result of the increased flexibility of the nucleosomal array and the potential to explore more entangled conformations.

**DNA twist-defect propagation is facilitated by H2A α2-helix bending.** We next move to characterize more nuanced changes in DNA conformation observed in our simulations. One of the most functionally interesting conformational transitions in nucleosomal DNA is the dynamics of twist-defects since it has a direct connection to nucleosome sliding[28]. Potential for local over-twisting and stretching of nucleosomal DNA was first observed in NCP X-ray structures. SHL ± 2 or ±5 can tolerate one insertion or deletion of a base pair resulting in NCP wrapping 145, 146, or 147 DNA base pairs within the same superhelical path (Supplementary Fig. 1). These variations were called twist-defects[58]. It was

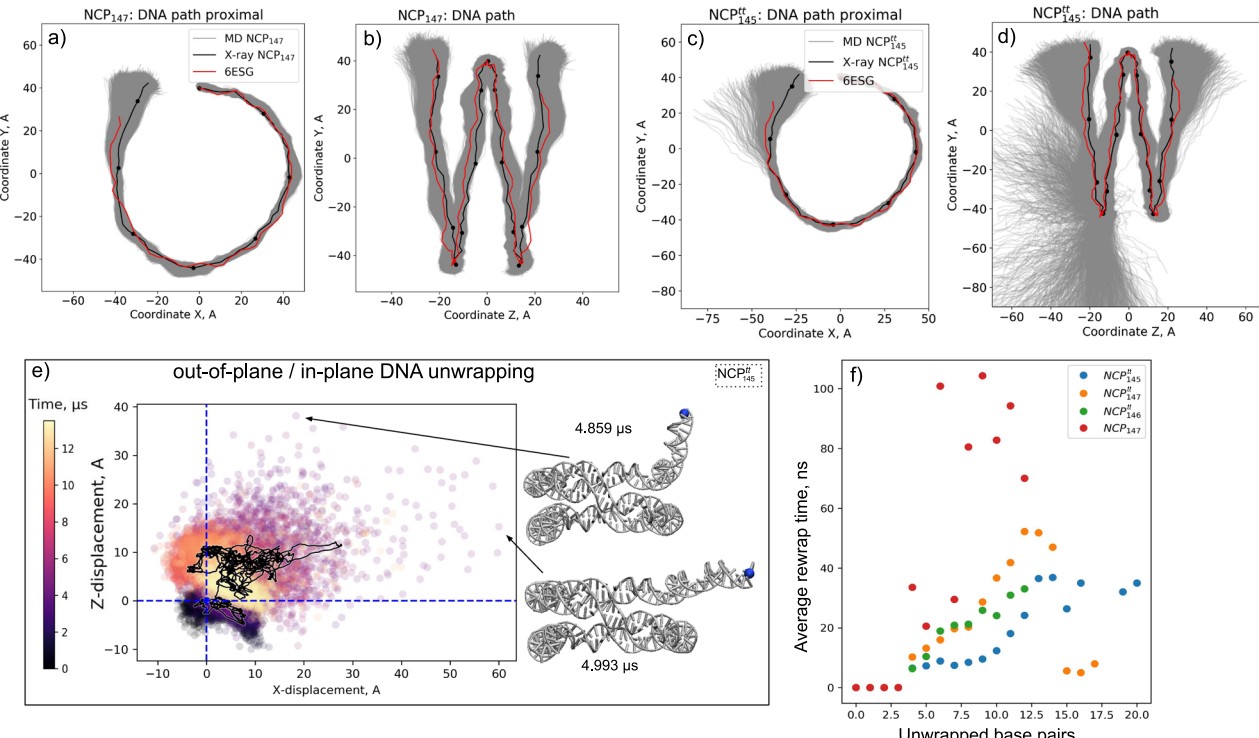

**Fig. 5 Characteristics of DNA unwrapping in nucleosomes. a–d** 2D projections of DNA paths in nucleosome reference frame for $NCP_{147}$ and $NCP^{tt}_{145}$ systems; **e** Scatter plot of DNA end fluctuations along the $Z$ and $X$-axis of the nucleosomal reference frame relative to its initial position in $NCP^{tt}_{145}$ simulation; **f** Average DNA rewrapping times as a function of DNA unwrapping extent estimated from MD trajectories. States with up to three unwrapped base pairs are considered as a wrapped state for this analysis.

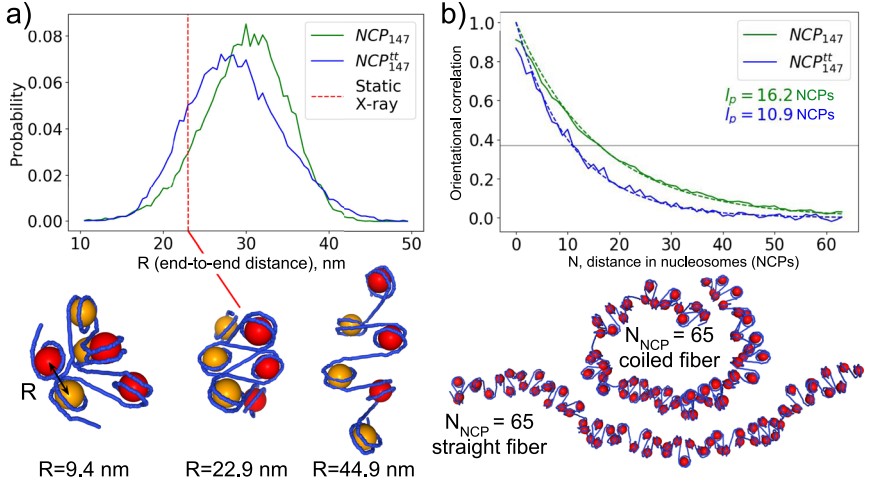

**Fig. 6 Effects of DNA unwrapping on chromatin fiber elasticity and bendability. a** End-to-end distance distribution in a simulated conformational ensemble of six NCPs connected by straight 15 bp linker DNA segments. Odd and even NCPs are colored in yellow and red, respectively, to highlight the fiber's "two-start" structure. **b** Decay of orientational correlations along the nucleosome fibers due to DNA unwrapping. Persistence length estimates are shown on the plot. Below the plot, two different conformations of the fiber are shown.

shown that nucleosome remodelers use such distortions for nucleosome sliding by moving nucleosomal DNA in a corkscrew inchworm fashion[21,24]. High-resolution cryo-EM studies revealed yet another unexpected intermediate present during the binding of nucleosome remodeling complexes at SHL ± 2 – the so-called half-twist-defects[24,28]. Half-twist-defects are characterized by local bulging of the tracking DNA strand around SHL ± 2 (this would be the top DNA strand in Fig. 2 if the remodeler binds at SHL -2). This bulging causes a specific asymmetric distortion of the DNA double helix all the way to the remodeler proximal end

of the nucleosomal DNA. The striking feature of the proposed distortion is that the tracking DNA strand shifts its register toward the bulge (5'–3' direction), while the nucleotides of the complementary strand remain almost in place.

To detect changes in DNA conformation during the simulations, we monitored the relative twisting of the double helix in the context of the nucleosome (Fig. 7a) and the changes in the positions of individual nucleotides along the reference path defined by the initial conformation of the DNA strands' backbones (Fig. 7b, Supplementary Fig. 10) (see Methods). The latter changes were

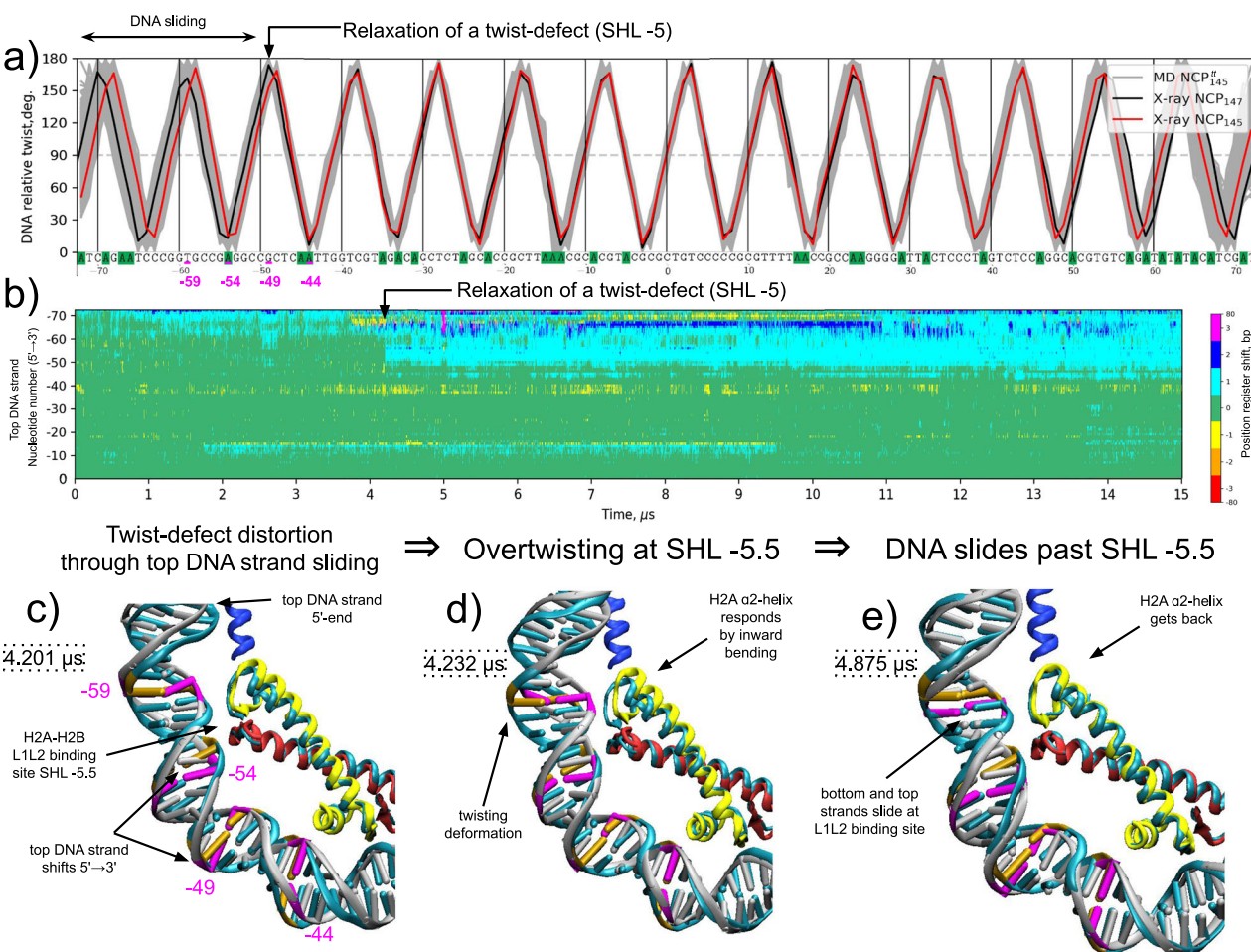

**Fig. 7 Formation/relaxation of twist-defects in NCP$^{tt}_{145}$ simulation. a** Plots of DNA relative twist profile along the DNA. The DNA sequence for the top strand of NCPtt145 is given along the X-axis. **b** A heatplot of changes in nucleotide positions for the proximal half of the top DNA strand during simulations. Starting from ~4 μs the half twist-defect relaxation causes the shift on the top DNA strand nucleotides in the region ~−50 to −72 by one step toward the dyad. **c–e** Successive stages of twist-defect relaxation at SHL -5 resulting in DNA sliding by 1 bp from SHL −7 to SHL −5. Snapshots are overlaid on the X-ray structure shown in cyan. Three base pairs (positions −59, −54, −49, and −44) around the SHL −5 region are highlighted in orange (X-ray positions) and magenta (MD positions). L1L2 binding sites are the DNA binding sites formed by L1 and L2 loops of histone folds. SHL stands for superhelix location.

characterized in discrete steps of "position register shift" reflecting how far away along the reference DNA strand conformation a particular nucleotide has shifted. Among the simulated systems, the NCP$^{tt}_{145}$ system was based on an X-ray structure that has DNA overtwisting and 1 bp stretching around SHLs ±5 with respect to the NCP$_{147}$/NCP$^{tt}_{147}$ structures. In this system, we were able to detect the relaxation of the twist-defect on the proximal side that happened within the first 5 μs of the simulation and persisted till the end of the trajectory (Fig. 7). The relative twist plots in Fig. 7a show that due to this transition, the system was able to sample the DNA twisting states inherent to the NCP$_{147}$ X-ray structure. The analysis of the positions of nucleotides in individual DNA strands revealed that the DNA strand segment (from the DNA entry into the NCP to around position -50) was effectively pulled in toward the DNA dyad at the moment around 4.2 μs (Fig. 7b, Supplementary Fig. 10).

Detailed visual analysis revealed the details of this transition. In a spontaneous distortion (with a timescale of fewer than 1 ns), the top DNA strand's nucleotides at SHL -5 start to slide toward the dyad (Fig. 7c). However, this sliding is constrained upstream around SHL -5.5 through the bottom DNA strand's interactions with the L1L2 binding site of the H2A-H2B dimer. It takes around 30 ns for this distortion to evolve and cause overtwisting of the DNA around SHL −5.5 (Fig. 7d). In response to this

overtwisting at the L1L2 site, the C-end of the H2A α2-helix, which forms this site, starts to bend inwards toward the center of the octamer. To some extent, this state resembles a half-twist-defect where one strand has already slid while the other is still held by the contacts with histones. It takes ∼400 ns for the overtwisting to resolve resulting in the sliding of both DNA strands at SHL −5.5 concomitantly with the return of the H2A α2-helix to its X-ray like position (Fig. 7e). The overall process happened during the state of rapid DNA breathing fluctuations (in the 0–15 bp range corresponding to unwrapping state 1 in Fig. 3), which loosens interactions with DNA upstream of SHL −5.5, likely facilitating twist defect relaxation.

The active participation of the H2A α2-helix in this process is further discussed below highlighting the importance of octamer plasticity in nucleosomal DNA dynamics.

**Plasticity of the histone octamer**. The alignment of MD snapshots to the common NRF (see Methods) allowed us to visualize and compare conformations and positions of protein backbone using 2D projections. The mapping of histone α2-helices shown in Fig. 8a, b and Supplementary Fig. 11 represents the changes in the overall geometry of the histone octamer for NCP$^{tt}_{145}$ and NCP$_{147}$, respectively. Below we focus on NCP$^{tt}_{145}$ analysis since it

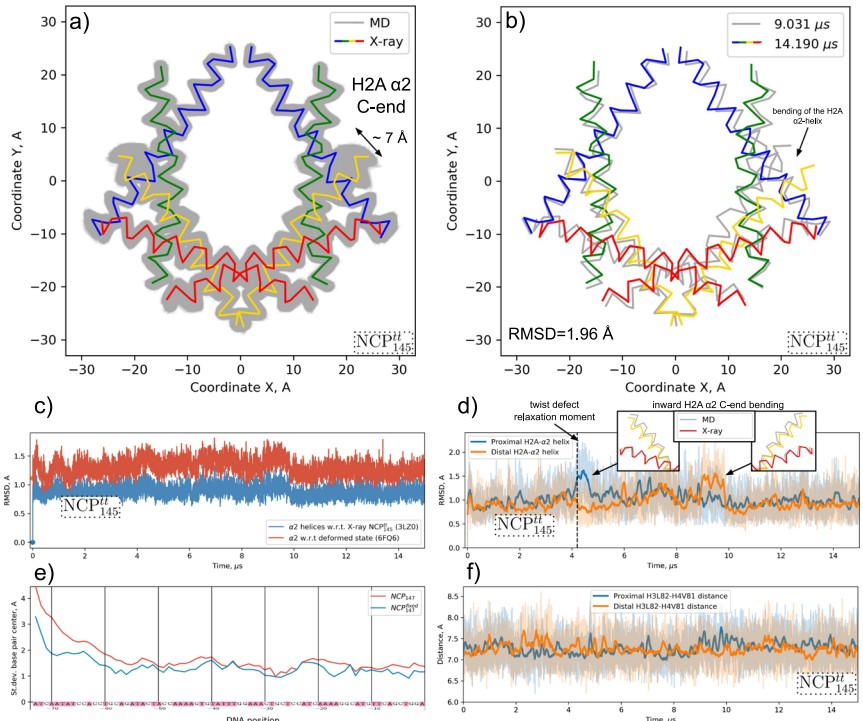

**Fig. 8 Plasticity of the histone octamer in NCP and its effects on the nucleosomal DNA dynamics. a–d** Present data for NCPtt145 simulations. **a** 2D projections of histone α2-helices' Cα-atoms (α2-Cα-atoms) on the plane perpendicular to the superhelical axis for MD snapshots vs. X-ray structure. **b** Same as (**a**), but for the two MD snapshots with the maximum RMSD as measured by positions of α2-Cα-atoms. **c** Variation of RMSD with simulation time measured for α2-Cα-atoms positions with respect to the initial X-ray structure and a cryo-EM structure of a "squeezed" NCP (PDB ID 6FQ6)[25]. **d** RMSD calculated for Cα-atoms of the H2A α2-helices as a function of simulation time. Insets show inward bending of the helix associated with higher average RMSD. **e** Average DNA fluctuations for one half of the nucleosomal DNA compared for NCP147 simulations (<8 μs) and NCPfixed147 simulations with restrained histone folds. **f** Distance between Cα-atoms of H3L82 and H3V81 residues. Thick lines in (**d**) and (**f**) show signal smoothed with Savitzky-Golay filter.

showed more diverse dynamics events (e.g., twist-defect relaxation). Although the RMSD of Cα-atoms with respect to the X-ray structure was on average below 1 Å (Fig. 8c), the maximum RMSD between the snapshots was 1.96 Å (Fig. 8b) and the distance between the different positions of individual Cα-atoms exceeded 5 Å. Such large fluctuations were observed for the C-end of the H2A α2-helix, which is located close to the L1L2 DNA binding site at SHL ± 5.5. These fluctuations were coupled to the bending of the helix. The RMSD plots for the Cα-atoms of the individual H2A α2-helices revealed conformational states with increased RMSD relative to the X-ray structure, which were accompanied by the bending of the α2-helices inward toward the center of the octamer and persisted for around 1 μs during the dynamics (Fig. 8d). As discussed above, this bending was found to be an essential process during twist-defect relaxation and DNA sliding. This suggests that overall octamer plasticity may also facilitate the nucleosomal DNA dynamics. To further test this hypothesis, we conducted a simulation with an artificially restrained histone folds Cα-atoms (NCPfixed147) to estimate the effects of the restraints on the positional fluctuations of the DNA along the sequence. As shown in Fig. 8e, restraining histone octamer plasticity indeed suppressed DNA fluctuations, especially around SHL ± 5 and ±2 (DNA positions around ±50 and around ±20). Moreover, the NCPfixed147 system showed a considerably smaller number of DNA unwrapping or breathing events of lower magnitude than its unrestrained counterpart (NCP147) during the available 8 μs of simulation time (Supplementary Fig. 2).

We also compared the octamer plasticity observed in simulations with the recently reported deformed (squeezed by 8% along the dyad axis) NCP structure seen in cryo-EM (PDB ID

6FQ6)[25]. On the 2D projection plots, the positions of the Cα-atoms in the deformed structure almost entirely fit into the cloud of positions sampled during MD simulations (Supplementary Fig. 12). RMSD analysis showed that certain NCP conformations during the MD simulations were closer to the deformed state than to the original canonical X-ray state (Fig. 8c). This at least suggests that experimentally observed perturbation of the nucleosome structure is on the same spatial scale as the fluctuations observed in our simulations.

The introduction of the H3L82C-H4V81C disulfide cross-links into the H3-H4 dimers was experimentally shown to inhibit thermal sliding of nucleosomes as well as ATP-dependent sliding by SNF2h remodeler, presumably by constraining octamer plasticity[22,25]. We monitored the distance between the Cα-atoms of this residue pair (Fig. 8f). Although no significant changes in the average distance were observed over the simulation time frame, the distance fluctuations ranged from 6.2 Å to 8.6 Å. The maximum distance between Cα-atoms of cross-linked cysteines if their side chains are in optimal orientation is around 7.5 Å. This already suggests that such disulfide cross-links are not compatible with all the conformational states observed in our dynamics simulations likely resulting in dynamical changes. These, however, await further characterization.

**DNA unwrapping is coupled to local DNA distortions near the dyad.** Formal analysis of changes in individual nucleotide positions also revealed that during the simulations, distortions in DNA were common close to the dyad around SHL ± 1 − ±1.5

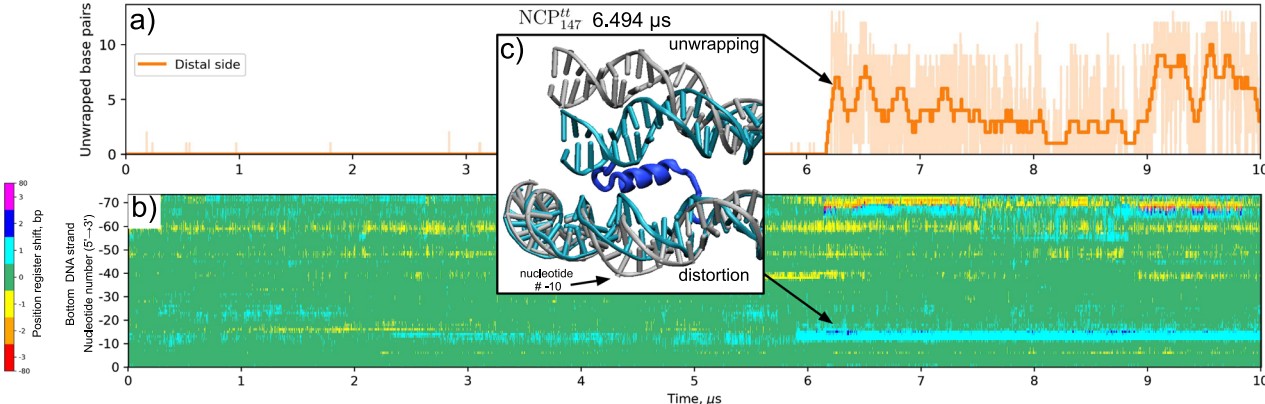

**Fig. 9 Coupling between DNA unwrapping and DNA loosening near the dyad through H3-latch interactions. a** DNA unwrapping as a function of simulation time of the distal DNA end in NCP$^{tt}_{147}$. **b** DNA distortions of the proximal half of the nucleosomal DNA visualized through the position register shift plot for the bottom DNA strand. **c** An MD snapshot showing the simultaneous DNA unwrapping, DNA distortion near the dyad, and detachment of the H3-latch from the inner DNA gyre near the dyad. Initial X-ray state is shown in cyan.

(Figs. 7b, 9b, Supplementary Fig. 10). Unlike the twist-defects, these distortions were localized to a stretch of 3-4 base pairs without affecting DNA regions upstream or downstream of the segment (Fig. 9c). The analysis of histone-DNA contacts (Supplementary Movie 7) and visual inspection of the trajectories revealed that the formation of these distortions followed the loss of H3-latch interactions with the distorted DNA segment. Particularly the loss of these interactions was accompanied by the removal of H3R40 from the DNA minor groove near the dyad. The destabilization of the H3-latch interactions with the DNA is in turn coupled to the unwrapping of the DNA ends, as discussed above. Simultaneous analysis of DNA unwrapping and distortion of the said region for NCP$^{tt}_{147}$ system is shown in Fig. 9. A clear synchronism in DNA unwrapping and formation of distortion near SHL 1 can be seen.

The results presented above suggest that DNA unwrapping promotes the loss of histone-DNA contacts near the dyad and distortion of the DNA near the dyad. This in turn, likely facilitates the sliding of nucleosomes via twist-defect propagation at the dyad region.

## Discussion

We performed equilibrium MD simulations of NCPs at physiological ionic conditions on a timescale of up to 15 μs. This long timescale enabled us to study in atomistic detail functionally relevant dynamical modes of nucleosomes such as breathing/unwrapping of nucleosomal DNA ends, relaxation/formation of twist defects within the DNA, conformational rearrangements of histone tails and the globular core of the histone octamer.

Unwrapping of the terminal helical turn of the nucleosomal DNA (which we call DNA breathing) was a common feature of NCP dynamics in our multi-microsecond simulations. This is consistent with force spectroscopy measurements showing little to no barrier for DNA unzipping in this region and ~0.2–0.6 equilibrium constant estimates for the initial stages of DNA unwrapping in FRET experiments[56,59]. We showed that this breathing is initiated by the rupture of the interactions between the DNA end and residues within the H3 39-49 region, which we termed "H3-latch". A similar chain of events was observed by Winogradoff and Aksimentiev in MD simulations of nucleosomes at high ionic strength (1 M MgCl$_2$ or 3 M NaCl)[37]. The comparison of the unwrapping dynamics during simulations suggests that lower ionic strength results in a higher frequency of DNA fluctuations once the stable H3-latch region contacts are

ruptured. This region is a known target of important epigenetic modifications, such as H3Y41, H3T45 phosphorylation, and H3R42 methylation[60]. Such modifications should affect either electrostatic interactions with the DNA end in the case of phosphorylation or the propensity to interact with the DNA minor groove for H3R42 methylation (H3R42 was observed interacting with the DNA minor groove in our simulations).

The unwrapping/rewrapping rate and motion range of the DNA ends was dramatically reduced by the interactions with the H3 N-tails and H2A C-tails in our simulations. This is in line with FRET and SAXS studies showing that NCPs with partially clipped H3-tails are less compact and their DNA termini are on average located further away from the histone core[52,61], as well as NMR studies showing that H3-tail robustly interacts with the nucleosome core[43]. Our simulations provide insights of high temporal and spatial resolution into the actual interplay between DNA dynamics and histone tails' dynamics during DNA breathing/unwrapping. Once the key interactions with the H3-latch are ruptured, the nucleosomal DNA ends per se tend to fluctuate rapidly on the 10–100 ns timescale with a high degree of in-plane and out-of-plane bending and conformational variability. However, the interactions with histone tails heavily modulate the magnitude of the fluctuations and the average DNA position during these fluctuations. It is the rearrangement of histone tails' conformation on the timescale of microseconds and beyond that actually governs the kinetics of DNA unwrapping/breathing.

The most advanced experimental methods to study nucleosomal DNA unwrapping dynamics currently rely on introducing a FRET pair of fluorescent labels at spatially close positions in nucleosome and measuring the dynamics of their fluorescence, which is affected by the distance between the labels due to Förster resonance energy transfer effect[20,56,57,62]. Measuring the fluctuations in fluorescence intensity usually requires photon flux accumulation in 1–3 ms intervals, limiting the temporal resolution of the method to the millisecond timescale. However, single-photon counting techniques, fluorescence correlation spectroscopy (FCS) approaches[57], fluorescence lifetime measurements[20] combined with advanced statistical analysis techniques allow to probe sub-millisecond to microsecond scale dynamics of nucleosomes. The interpretation of these measurements also requires careful consideration of potential spurious contributions from the photophysical effects and understanding the relation of the observed FRET efficiency variations to the actual magnitude of the distance fluctuations between the label attachment sites.

Gansen et al. have detected relaxation times of conformational transitions in the histone core as fast as 14 µs[20]. While their label positions did not allow to directly probe DNA breathing, they have revealed additional relaxation components in histone core dynamics between 1 and 100 µs, which they suggested to be due to DNA breathing. Wei et al. employing a hybrid FCS-FRET technique with one of the labels directly attached to the nucleosomal DNA end, reported detectable DNA breathing times on the scale of 1–10 ms[57]. They have also detected microsecond components in the dynamics of the signal, but these were attributed to fluorophore photophysics effects. The results of these and other studies suggest that spontaneous unwrapping of several DNA turns likely requires hundreds of milliseconds[19], while small scale DNA breathing happens at the sub-millisecond to microsecond timescale. Our estimates suggest that in NCP simulations with full-length histone tails, the magnitude of the observed DNA breathing motions at the 15 µs timescale should affect FRET efficiency between a pair of fluorescent labels attached to the DNA end and a position near the dyad (Supplementary Figs. 8, 9). The variation in the signal should have relaxation components in the microsecond range. The observed DNA fluctuations in our simulations likely represent the initial stages of a slower process with a higher displacement magnitude of the DNA ends, which is promoted by further slow rearrangement of histone tails' conformations.

We have shown that the amount of DNA breathing observed in our MD simulations in NCP system with full-length histone tails is already sufficient to provide considerable variability to the conformation of chromatin fibers composed of NCPs connected by linker DNA segments. This puts forward nucleosomal DNA breathing/unwrapping as an important factor affecting chromatin structure plasticity at the supranucleosomal level and beyond. The chromatin fiber dynamical variability is likely important for forming higher-order chromatin structure through weak interactions arising due to liquid-liquid phase separation, DNA supercoiling, and macromolecular crowding[63]. With high-resolution experimental data on the positions and contacts of the individual nucleosomes in the genome becoming available through a number of methods such as Micro-C[64,65], we envision the importance of accounting for DNA breathing in reconstructing chromatin structure with high resolution from these data.

We observed that unwrapping of the nucleosomal DNA ends was coupled to the destabilization of the DNA near the dyad through their mutual interactions with the H3-latch. The observed effect raises an interesting possibility of DNA unwrapping promoting nucleosome sliding by poising the DNA near the dyad to the easier passage of DNA twist-defects. Interestingly some remodeling complexes that move nucleosome along the DNA, such as Chd1, engage with nucleosomal DNA ends and unwrap them upon binding[66].

There has been a lot of progress recently in understanding how ATP-dependent remodelers translocate nucleosomes along the DNA through DNA twist-defect propagation mechanisms in a corkscrew inchworm-like fashion[21,24]. Their ATPase subunits (e.g., Snf2, SNF2h, etc.) bind at SHL ± 2 and swing between open and closed states through successive cycles of ATP-hydrolysis. These conformational changes cause the DNA to be first pulled in from the remodeler proximal side of the nucleosome before it is pushed out from the other side. Cryo-EM studies suggest that the first intermediate in this cycle has a remarkably distorted DNA conformation, where one of the DNA strands (the tracking strand) is pulled in more than the other DNA strand (the guide strand) causing one DNA strand to slide past the other by 1 bp[24]. The latter deformation has been termed a half-twist-defect[28]. Our detailed analysis of DNA-histone interactions in nucleosomes

revealed that for each half of the NCP one DNA strand has a considerably fewer number of dynamically stable contacts with the histone octamer than the other strand. The former strand exactly corresponds to the one which is pulled in in the context of the nucleosome-remodeler complexes discussed above. These findings suggest that the mechanisms of ATP-dependent DNA translocation via half-twist-defect formation are facilitated by the DNA strand asymmetry of histone-DNA interactions in the nucleosome.

Nucleosomes are also known to translocate spontaneously in a thermally assisted diffusion process. How and to what extent twist-defects contribute to this process remains elusive. X-ray structures have shown that depending on the DNA sequence nucleosomal DNA conformation can be stretched/compressed by 1 bp around SHL ± 2 or ±5 due to twist-defects. Further footprinting studies suggested that in solution NCPs exist as a mixture of different twist-defect states, while in X-ray studies crystal environment stabilizes certain conformations[58]. In our simulations of NCP based on 601 high affinity DNA sequence, we were able to directly observe the relaxation of DNA overstretching around SHL -5 resulting in sliding of the terminal 2.5 helical turns of the nucleosomal DNA along the histone octamer. While the breathing of the DNA ends likely contributed to this process, the relaxation happened within the wrapped portion of the nucleosomal DNA. In this process, the top DNA strand (which has fewer stable contacts with the octamer) was spontaneously pulled in toward the center of the NCP causing partial relaxation of the accompanying DNA overtwisting at SHL -5 and generation of an additional distortion at SHL -5.5. This state resembled a half-twist-defect state where one of the DNA strands showed higher displacement than the other. The distortion at SHL -5.5 was facilitated by the considerable inward bending of the C-end of the H2A α2-helix. The resolution of this distortion intermediate resulted in DNA sliding toward the dyad and reestablishment of an X-ray like position by the H2A α2-helix. The observed mechanism provides direct evidence for the importance of histone octamer plasticity in DNA translocation around nucleosomes (see Methods).

The observed spontaneous sliding of the two terminal turns of the nucleosomal DNA suggests further potential pathways of DNA translocation around the NCP. In addition to SHLs ±5, structurally similar positions that may harbor twist-defects exist around SHLs ±2. The overall DNA translocation around the histone octamer may result due to the dynamic exchange of twist-defects between these sites.

The globular core of the histone octamer has been regarded for some time as a rigid modular structure assembled from H2A-H2B dimers attached to the H3-H4 tetramer. Recent studies, however, have put forward the importance of histone octamer plasticity and dynamics even at the level of the intrinsic dynamics of individual histone dimers. It has been reported that the introduction of cross-links inside individual histone dimers may impede spontaneous and ATP-dependent nucleosome sliding[22,25] and compaction of nucleosomal arrays by heterochromatin protein 1[27]. Asymmetric deformations of the octamer have been suggested to be implicated in the allosteric communication between the two faces of the nucleosome impeding simultaneous binding of SNF2h remodeler protomers to both sides of the NCP[30]. Several cryo-EM structures of NCP conformations with deformed octamer have been reported suggesting coupling between DNA and overall deformation of the histone octamer, slight reorientation of the histone α-helices[31]. However, the exact dynamical pathways of these processes remain elusive, especially at the atomistic level. To track octamer plasticity in our simulations, we have analyzed the variation in positions of the histone fold α-helices—the key elements which define octamer structure.

We show that on a multi-microsecond timescale, the most different conformations of the octamer have their histone fold Cα-atoms displaced on average by around 2 Å. The average magnitude of these deformations is on the same scale as that seen in cryo-EM structures of the alternative canonical nucleosome substates[25,31]. The octamer elements where we have observed maximum fluctuations reaching 7 Å were the C-ends of the H2A α2-helices. These motions resembled bending of the C-ends of the helix toward and outwards from the center of the octamer. Similar outward H2A α2-helix bending was observed in some cryo-EM structure of NCPs with partially unwrapped DNA[31]. Our data suggest that inward bending of the helix is also a common conformation observed in dynamics. As suggested by our simulations, this inward bending has a functional role in allowing the DNA to slide past the L1L2 H2A-H2B binding site (at SHL ± 5.5) during twist-defect relaxation. Here we can formulate an experimentally testable prediction that restricting H2A α2-helix mobility by introducing a site-specific cross-link between the C-end of the H2A α2-helix and N-end of the H2B α2-helix will impede nucleosome sliding. Interestingly, a cross-link H3L82-H4V81 in a structurally related position in the H3-H4 dimer has already been shown to impede thermally assisted and ATP-dependent nucleosome sliding[22,25]. While in our simulations (potentially due to limited simulations time), we did not observe considerable H4 α2-helix bending, the structural and positional similarity of the H2A-H2B and H3-H4 dimers suggests that a similar flexibility mechanism may exist within the H3–H4 dimer. If such a mechanism exists, it may serve as an explanation for the above experimental findings.

Taken together, the results of our study provide further mechanistic insights into the role of histone dynamics in nucleosome unwrapping, sliding, and supranucleosomal organization of chromatin. This in turn paves the way toward understanding the effects of various epigenetic alterations of the nucleosomes (such as histone PTMs, histone variants, DNA methylation, interactions with chromatin proteins, etc.) on genome functioning.

## Methods

**Molecular dynamics simulations**. Simulations were performed using GPU-accelerated GROMACS 2018.1[67] with AMBER ff14SB force field[68] supplemented with parmbsc1[69] DNA and CUFIX[70] ion parameter corrections. The latter corrections adjust nonbonded interactions mainly between charged atom groups (e.g., solvent ions, DNA phosphate oxygen atoms, positively charged nitrogen atoms in proteins) and were shown to improve the description of both biological electrolyte solutions[70] and disordered protein conformations[71]. Both aspects are essential for modeling nucleosomes, which are highly charged and contain disordered histone tails. TIP3P water model was used[72]. Initial coordinates were derived from corresponding structure files in the PDB database (Table 1). For NCP$_{147}$ simulation, the histone tails' conformations for the distal half of the NCP (chains E, F, G, H) were adjusted to match the conformation of their proximal symmetry-related mates (chains A, B, C, D). For simulations with truncated histone tails, the truncation sites are depicted in Fig. 1e. Truncation sites were chosen to remove the histone tails' flexible parts while still retaining some parts near the globular core that make stable contacts with the DNA or histones (e.g., H3 and H2B N-tails protruding between the DNA gyres or H2A C-tail interacting with H3-H4 dimer). For simulations with fixed histone folds, the Cα-atoms of α1, α2, and α3-helices in all histones were restrained to their original positions using a harmonic potential with force constant of 1000 kJ mol$^{-1}$ nm$^{-2}$. To suppress potential fraying of terminal DNA base pairs during simulations, the distance between glycosidic nitrogen atoms (N1 for thymine or cytosine and N9 for adenine or guanine) of the terminal base pairs was constrained using a harmonic potential with force constant of 1000 kJ mol$^{-1}$ nm$^{-2}$. All systems were placed in a truncated octahedron simulation box with periodic boundary conditions set at least 2 nm away from the NCP atoms. Next, water molecules were added to the box for solvation, Na and Cl ions were added to neutralize the charge and bring the ionic strength to 150 mM (Fig. 1b). Minimization was performed via the steepest descent gradient method for 10,000 steps with positional restraints on heavy atoms of 500 kJ mol$^{-1}$ nm$^{-2}$. Then the system was equilibrated in five consecutive steps with gradual reduction of restraining potential: 1. 100 ps with positional restraints of 500 kJ mol$^{-1}$ nm$^{-2}$ with 0.5 fs time step; 2. 200 ps with positional restraints of 50 kJ mol$^{-1}$ nm$^{-2}$ with 2 fs time step (and further the same); 3. 200 ps with positional restraints

of 5 kJ mol$^{-1}$ nm$^{-2}$; 4. 200 ps with positional restraints of 0.5 kJ mol$^{-1}$ nm$^{-2}$; 5. 200 ps of unrestrained simulations. Temperature was maintained at 300 K using velocity rescale scheme[73] and pressure at 1 bar using Parrinello-Rahman barostat[74]. Verlet non-bonded cut-off scheme with grid neighbor search algorithm updated every 10 steps and 0.8 nm cut-off radius for van der Waals interactions with dispersion correction for energy and pressure was used. Particle Mesh Ewald (0.8 nm real space cut-off, fourth-degree PME order, 0.12 nm Fourier spacing) approach was used to account for Coulomb interactions. All bonds were constrained using the fourth-order LINCS algorithm with one iteration used to correct for rotational lengthening. During MD production runs, the integration step of 2 fs was used, trajectory frames were saved every 1 ns. Detailed simulated systems' description is provided in Supplementary Note 1. System details including the number of atoms, water molecules and box sizes are provided in Supplementary Table 1. Production run protocol files may be found at https://doi.org/10.5281/zenodo.4590559.

Simulations were performed in parallel on the Lomonosov-2 supercomputer[75] using 8 computing nodes, each having 14 CPU cores and one NVidia Tesla K40 GPU. The simulations progressed at an average speed of 60 ns per day. Two 7-thread MPI tasks were assigned to every node. GPUs were used to calculate nonbonded interactions.

**Analysis**. Custom analysis programs and pipelines were written in Python 3, integrating the functionality of GROMACS (trajectory preprocessing)[76], MDAnalysis (coordinate manipulation, 3D alignment)[77], VMD (visualization)[78], and 3DNA (determination of DNA base pair centers, calculation of base pair and base pair step parameters)[79].

Two vectors were instrumental for analyzing nucleosome geometry: the superhelical axis vector (**OZ**, calculated from the DNA path)[41] and the dyad twofold pseudo-symmetry axis vector (**OY**, defined as the perpendicular connecting **OZ** with the center of the central DNA base pair). Together with the **OX** vector defined as **OY** and **OZ**'s cross product, the three vectors formed the coordinate system of the NRF (Fig. 1a). NRF was determined for the NCP X-ray structure (PDB ID 1KX5). All other structures and conformations were aligned to it by minimizing the root-mean-square deviation (RMSD) between the Cα-atoms of the histone folds α-helices (α1, α2, α3). NRF was used to visualize molecular geometry in various projections and calculate DNA relative twist (local twist)[80]. Any RMSD values reported in this study were calculated by first aligning the respective structures/conformations to the NRF and then computing RMSD without an additional round of pairwise alignment.

Relative twist describes the rotation of the DNA with respect to the histone octamer's surface. It may be used to track the rotational positioning of the DNA and its twist-defects (see Supplementary Fig. 13 for additional details). For every nucleotide pair, the base pair orientation vector (BPV) was calculated as C1'-C1' (ribose C-atoms, forming N-glycosidic bonds) vector from top to bottom DNA strand. The radial vector in the cylindrical coordinate system (ORxy) was defined as the perpendicular pointing from the nucleosome superhelical axis (OZ) to the base-pair center (BPV center). Relative twist was defined as the angle between the OXY plane (the front plane in NRF coordinates) and the projection of the BPV onto the plane formed by two vectors OZ and ORxy. In such a definition, the relative twist parameter varies in the range from −180 to 180 degrees. For easier perception we plot the absolute value of the parameter, which results in a continuous curve.

Atom-atom contacts between histones and DNA were calculated as non-hydrogen atom pairs at the distance of <4 Å. Contact between an amino acid residue and a nucleotide was considered to be present if any of their atoms were in contact. Stable amino acid residue-nucleotide contacts were defined as contacts present in at least 90% of MD frames during the given time period.

The extent of DNA unwrapping was based on the analysis of either DNA geometry or histone-DNA contacts. In the first case, a DNA segment (starting from either end of the nucleosomal DNA) was considered to be unwrapped from the NCP if the position of every base pair in that segment (as provided by 3DNA) was more than 7 Å away from the initial position of any base pair in the X-ray structure. In the second case, a DNA segment was considered unwrapped if there were no atom-atom contacts between that segment and the histone octamer's globular core (defined as non-tail residues in Fig. 1e).

DNA twist-defects and DNA geometry distortions were characterized by calculating relative twist profiles and DNA register shift profiles. Unlike relative twist, which characterizes the rotation of an individual base pair with respect to the octamer surface, DNA position register shift characterizes the position of an individual nucleotide in a given DNA strand. To calculate the register shift for a given nucleotide, it was matched to the closest nucleotide in the initial X-ray structure by comparing the positions of their C1' atoms after NRF alignment. A match between different nucleotides indicated a register shift, positive for the shift toward 3'-end and negative otherwise.

**Nucleosome structural elements**. The main structural elements are shown in the Fig. 1e. Helices are defined by taking the minimum length of alpha-helices over symmetric chains in 1KX5 X-ray structure. For alignment, C-alpha atoms of the histone folds (which are defined as α1, α2, and α3-helices as in the figure) were used. Further in our analysis, we often analyzed the contacts between histone core

and histone tails separately. The definition of histone tails is provided in Fig. 1e, histone globular core is defined as all other parts of the histone sequence.

**Chromatin fiber simulations**. MD based models of chromatin fibers were generated by connecting random NCP snapshots from MD trajectories by 15 bp straight linker segments of B-DNA. To avoid potential end effects, the last three base pairs of NCP DNA were also replaced by straight B-DNA segments. Thus, the position of sites ±70 base pairs from the dyad actually determined the linker segments' orientation. Every NCP was represented by a sphere 12 nm in diameter and every DNA base pair by a 2 nm sphere through a coarse-graining procedure. The generated coarse-grained fiber models were discarded if the steric overlap between NCPs or linker DNA was present. To obtain a statistical sampling of the fiber geometry around 10,000 conformations were generated for every fiber model. Modeling was implemented through a combination of 3DNA[79] and custom Python scripts. The decay of orientational memory (correlation) between the starting nucleosome and a selected downstream nucleosome was calculated as the average cosine of the angle between the vector specifying that nucleosome's orientations in the simulated ensemble and the vector specifying its reference orientation in an ideal fiber based on X-ray NCP structures.

**Reporting summary**. Further information on research design is available in the Nature Research Reporting Summary linked to this article.

## Data availability
Data supporting the findings of this paper are available from the corresponding author upon reasonable request. A reporting summary for this Article is available as a Supplementary Information file. Trajectories and protocols are available for preview and download from GitHub at https://intbio.github.io/Armeev_et_al_2021 and are archived via Zenodo at https://doi.org/10.5281/zenodo.4590559.

## Code availability
Computer code is available from GitHub under https://github.com/intbio/Armeev_et_al_2021 and is archived via Zenodo https://doi.org/10.5281/zenodo.4590559.

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

## Acknowledgements

We thank Prof. Vladimir V. Palyulin for valuable discussions. This research was supported by Russian Science Foundation grant #18-74-10006 (MD simulations and analysis), Russian Foundation for Basic Research grants #20-34-70039 (supranucleosome structure modeling), #19-34-51053 (development of protein-DNA analysis algorithms) and by the Interdisciplinary Scientific and Educational School of Moscow University "Molecular Technologies of the Living Systems and Synthetic Biology". A.K.S is supported by HSE University Basic Research Program. The research was carried out using the equipment of the shared research facilities of HPC computing resources at Lomonosov Moscow State University.

## Author contributions

A.K.S. conceived the project, designed simulations and data analysis algorithms. G.A.A., A.S.K., and A.K.S. performed simulations, developed and applied analysis algorithms. M.P.K. and G.A.K. contributed to the study design and paper preparation. G.A.A. developed algorithms and performed chromatin fiber simulations. A.K.S wrote the paper with support from all authors.

## Competing interests

The authors declare no competing interests.
