## [Peer Review File · Nature Communications]

Reviewer #1 (Remarks to the Author):

This spectacular manuscript describes the most comprehensive all-atom molecular dynamics study of nucleosome particles performed to date. Using a state-of-the-art molecular force field model, the authors carried out multiple fully atomistic, explicit solvent MD simulations of several nucleosome particle variants, reaching the simulation time scale of 15 microseconds. At that simulation timescale, previously unseen effects are coming to light. First and foremost, this extended timescale, combined with the use of CUFIX corrections, allowed the disordered histone tails to explore a large ensemble of possible molecular configurations, which makes this study the first to achieve that. Without histone tails, DNA was observed to spontaneously unbind from the histone core, but, in the presence of the tails, the unbinding was largely subdued. Analysis of the simulation trajectories identified molecular interactions responsible for DNA binding/rebinding, examined formation and propagation of twist defects, plasticity of the core particle and, finally, the correlation between DNA unbinding and structural changes in the core proteins. Impressively, the authors also constructed a simple yet effective model of a nucleosome fiber, showing that the DNA unbinding observed in the all-atom simulations is sufficient to reproduce the heterogeneity of the nucleosome fiber structures.

Overall, this is a landmark computational work that pushes the envelope for the MD field. Specific to the nucleosome field, the study provides new insights into the mechanics of nucleosome re-positioning and offers microscopic interpretations to experimental results. The manuscript is clearly written. The figures and animations are of superb quality. I strongly recommend this manuscript for publication in Nature Communication.

The authors are asked to address the following minor points.

Methods: Please specify the number of atoms in each simulations system.

Modeling conformational dynamics of full-length histone tails has been difficult because of the insufficient simulation time scale and imperfections of the molecular force field, which generally predict disordered proteins to adopt much more compact conformations than those observed in experiment. Such force field limitations could have been a concern for standard AMBER or CHARMM simulations. But the authors used CUFIX, which was shown to improve the description of disordered protein conformations [J. Phys. Chem. Lett. 7, 3812]. The authors are asked to comment on that, perhaps in the methods section or in the discussion, to pre-empt possible concerns of the readers not familiar with the effect of CUFIX on disordered protein simulations.

Among the residues that the authors find to form long lasting bonds with the DNA, are there any that are evolutionary conserved?

Line 254 and in several places in the manuscript. The authors use the word “Evolution” to describe temporal behavior, but “evolution” has a more narrow meaning – deterministic improvement with time. Consider rewriting, for example, “Figure 2 shows how changes as a function of simulations time”

The authors describe a detailed chain of events associated with spontaneous unbinding and rebinding of DNA to the histone core. A very similar unbinding/rebinding was reported in Ref 37. Is the mechanism of the unbinding/rebinding transition the same in the two studies?

Line 437, Figure 8d should probably be 7d.

The description of Figure 7 and Figure 8 in the main text does not call the figure panels in the alphabetical order (a, b, c, etc). Consider rearranging the panel to follow the main text narrative.

Reviewer #2 (Remarks to the Author):

I am not a specialist in all-atom MD simulations but have worked a great deal on nucleosomes, using much more coarse-grained models. I have read all the papers on MD simulations of nucleosomes but found that the insights I could gain from the all-atom studies were rather limited. This has changed with this excellent study where the authors have reached long enough time-scales (15 microseconds) so that interesting dynamics can be observed in the nucleosome model.

A few examples:

- The plots (and movies) with the histone-DNA contacts are extremely useful, e.g. to build again more coarse-grained models.
- The observation of different unwrapping states triggered by the the H3 latch is surprising and well explained in the paper.
- The impact of H2A alpha2-helix bending on twist propagation highlights (as also the previous point) the important role of the histone proteins on nucleosome dynamics, something that has long been underestimated. Twist defect diffusion is a crucial dynamic mode employed by chromatin remodelers.

The writing is very clear and the authors relate to the relevant literature. The web interface provided is also very useful. I very highly recommend to publish this excellent work with some very minor modifications:

1. line 226: typo, it should read "-32"
2. line 393: typo? Should it not read "missing base pair" instead of "additional base pair"?
2. NCP_145 shows highly asymmetric breathing (see Figure 3). Also various experiments with 601 nucleosomes have found this (e.g. J. D. Anderson and J. Widom, *J. Mol. Biol.* 296:979 (2000); A. W. Mauney, ..., L. Pollack, *Biophys. J.* 115:773 (2018)). It would be interesting to know whether the current study finds that the breathing occurs from the same side as in the experiments.
4. The years of publication are missing in the references.

Helmut Schiessel

Response to reviewers' comments

RE: Manuscript “Histone dynamics mediate DNA unwrapping and sliding in nucleosomes” by Armeev et al.

Original comments are highlighted in **bold blue**, authors answers in black normal font, changes to the manuscript in “*green italics*”.

Reviewer #1.

This spectacular manuscript describes the most comprehensive all-atom molecular dynamics study of nucleosome particles performed to date. Using a state-of-the-art molecular force field model, the authors carried out multiple fully atomistic, explicit solvent MD simulations of several nucleosome particle variants, reaching the simulation time scale of 15 microseconds. At that simulation timescale, previously unseen effects are coming to light. First and foremost, this extended times scale, combined with the use of CUFIX corrections, allowed the disordered histone tails to explore a large ensemble of possible molecular configurations, which makes this study the first to achieve that. Without histone tails, DNA was observed to spontaneously unbind from the histone core, but, in the presence of the tails, the unbinding was largely subdued. Analysis of the simulation trajectories identified molecular interactions responsible for DNA binding/rebinding, examined formation and propagation of twist defects, plasticity of the core particle and, finally, the correlation between DNA unbinding and structural changes in the core proteins. Impressively, the authors also constructed a simple yet effective model of a nucleosome fiber, showing that the DNA unbinding observed in the all-atom simulations is sufficient to reproduce the heterogeneity of the nucleosome fiber structures.

Overall, this is a landmark computational work that pushes the envelope for the MD field. Specific to the nucleosome field, the study provides new insights into the mechanics of nucleosome re-positioning and offers microscopic interpretations to experimental results. The manuscript is clearly written. The figures and animations are of superb quality. I strongly recommend this manuscript for publication in Nature Communication.

We would like to thank the referee for the overall strongly positive assessment of our study.

1 The authors are asked to address the following minor points.

Methods: Please specify the number of atoms in each simulations system.

We now explicitly refer in the main text to the Supplementary Table 1 which provides detailed information about the systems - number of atoms, water molecules, ions, box dimensions, etc.

Following text was added to the methods section:

“System details including the number of atoms, water molecules and box sizes are provided in Supplementary Methods and Supplementary Table ST1”

2 Modeling conformational dynamics of full-length histone tails has been difficult because of the insufficient simulation time scale and imperfections of the molecular force field, which generally predict disordered proteins to adopt much more compact conformations than those observed in experiment. Such force field limitations could have been a concern for standard AMBER or CHARMM simulations. But the authors used CUFIX, which was shown to improve the description of disordered protein conformations [J. Phys. Chem. Lett. 7, 3812]. The authors are asked to comment on that, perhaps in the methods section of in the discussion, to pre-empt possible concerns of the readers not familiar with the effect of CUFIX on disordered protein simulations.

We thank the referee for raising this point, a corresponding comment and a reference were added to the methods section.

“The latter corrections adjust non-bonded interactions mainly between charged atom groups (e.g., solvent ions, DNA phosphate oxygens, positively charged nitrogens in proteins) and were shown to improve the description of both biological electrolyte solutions⁷⁰ and disordered protein conformations⁷¹. Both aspects are essential for modeling nucleosomes which are highly charged and contain disordered histone tails.”

3 Among the residues that the authors find to form long lasting bonds with the DNA, are there any that are evolutionary conserved?

We thank the referee for raising this question. The problem of applying the common techniques of conservation analysis to the histones is that histones are ones of the most conserved proteins. For example, for H3 histone one may see this form multiple sequence alignments maintained by us at the Histone Database web site https://histdb.intbio.org/type/H3/variant/canonical_H3 . The core part of the canonical histones including the residues making contacts are almost universally conserved among a wide variety of species. In our opinion an interesting question of investigation is the substitution of some of the key amino acids of the H3-latch in certain divergent species or histone variants (e.g. in Trypanosoma H3P38Q, H3H40R, H3Y41W) and how they might affect DNA dynamics. However, we would prefer to retain these questions for a separate study.

4 Line 254 and in several places in the manuscript. The authors use the word “Evolution” to describe temporal behavior, but “evolution” has a more narrow meaning –

deterministic improvement with time. Consider rewriting, for example, “Figure 2 shows how changes as a function of simulations time”

We agree that in this manuscript the word evolution may be interpreted ambiguously. Corresponding changes were implemented throughout the manuscript.

5 The authors describe a detailed chain of events associated with spontaneous unbinding and rebinding of DNA to the histone core. A very similar unbinding/rebinding was reported in Ref 37. Is the mechanism of the unbinding/rebinding transition the same in the two studies?

We thank the referee for bringing this to our attention. Indeed, there are notable similarities between the mechanisms observed in our simulations and in Ref. 37. Particularly, both studies show that the residues of the H3-latch H3H39, H3R40, H3R49 form key contacts with the DNA that should be broken in order for the DNA unwrapping to start. Although, in terms of the details of the kinetics of DNA unwrapping/rewrapping we think there are certain differences attributable to different ionic conditions used in the two studies. Particularly, we see rapid fluctuations between wrapped and unwrapped conformations on the timescale of 20-50 ns (at least when terminal 10-15 bp are detached). We think that lower ionic strength stimulates DNA rewrapping once it is unwrapped due to thermal fluctuations.

We added the following comment to the text to discuss the similarity between the studies.

“A similar chain of events was observed by Winogradoff and Aksimentiev in MD simulations of nucleosomes at high ionic strength (1M MgCl₂ or 3M NaCl).³⁷ The comparison of the unwrapping dynamics during simulations suggests that lower ionic strength results in a higher frequency of DNA fluctuations once the stable H3-latch region contacts are ruptured.”

6 Line 437, Figure 8d should probably be 7d.

We thank the reviewer for the note. Fig. 8d shows the interplay between the H2A α 2-helix location and twist-defect propagation. However, we admit that mentioning it in the text was unclear in that section. We reformulated it.

“The active participation of the H2A α 2-helix in this process is further discussed below highlighting the importance of octamer plasticity in nucleosomal DNA dynamics.”

7 The description of Figure 7 and Figure 8 in the main text does not call the figure panels in the alphabetical order (a, b, c, etc). Consider rearranging the panel to follow the main text narrative.

We rearranged figures 7 and 8 to match the text in the alphabetical order.

Reviewer #2.

I am not a specialist in all-atom MD simulations but have worked a great deal on nucleosomes, using much more coarse-grained models. I have read all the papers on MD simulations of nucleosomes but found that the insights I could gain from the all-atom studies were rather limited. This has changed with this excellent study where the authors have reached long enough time-scales (15 microseconds) so that interesting dynamics can be observed in the nucleosome model.

A few examples:

- The plots (and movies) with the histone-DNA contacts are extremely useful, e.g. to build again more coarse-grained models.

- The observation of different unwrapping states triggered by the the H3 latch is surprising and well explained in the paper.

- The impact of H2A alpha2-helix bending on twist propagation highlights (as also the previous point) the important role of the histone proteins on nucleosome dynamics, something that has long been underestimated. Twist defect diffusion is a crucial dynamic mode employed by chromatin remodelers.

The writing is very clear and the authors relate to the relevant literature. The web interface provided is also very useful. I very highly recommend to publish this excellent work with some very minor modifications:

We thank the referee for a highly positive feedback on our manuscript

1. line 226: typo, it should read “-32”

We agree that such wording looks cumbersome and replaced the text with "(... *positions from -73 to -32 in ...*)"

2. line 393: typo? Should it not read “missing base pair” instead of “additional base pair”?

We tried to reformulate it in a more clear way.

“SHL ± 2 or ± 5 can tolerate one insertion or deletion of a base pair resulting in NCP wrapping 145, 146, or 147 DNA base pairs within the same superhelical path (Supplementary Figure 1)”

2. NCP_145 shows highly asymmetric breathing (see Figure 3). Also various experiments with 601 nucleosomes have found this (e.g. J. D. Anderson and J. Widom, J. Mol. Biol. 296:979 (2000); A. W. Mauney, ..., L. Pollack, Biophys. J. 115:773 (2018)). It would be interesting to know whether the current study finds that the breathing occurs from the same side as in the experiments.

The 601-sequence to our knowledge indeed has been shown to unwrap asymmetrically, this correlates with the number of TA-dinucleotides in the sequences (see Supplementary Figure 1b). Left side is a “strong” side and right side is a “weak” one. This is exactly what we see in our simulations. However, we are not confident that it is not a coincidence. Particularly, because the TA-dinucleotides that provide DNA kinking are localised at the inner DNA turn, while in our simulations only unwrapping of the outer part of the DNA is seen.

4. The years of publication are missing in the references.

We thank the reviewer for noticing this, we fixed that issue in the revised manuscript.